# The Top 100 Most Cited Scientific Papers in the Public, Environmental & Occupational Health Category of Web of Science: A Bibliometric and Visualized Analysis

**DOI:** 10.3390/ijerph19159645

**Published:** 2022-08-05

**Authors:** Vicenç Hernández-González, Josep Maria Carné-Torrent, Carme Jové-Deltell, Álvaro Pano-Rodríguez, Joaquin Reverter-Masia

**Affiliations:** 1Human Movement Research Group (RGHM), University of Lleida, Plaça de Víctor Siurana, 25003 Lleida, Spain; 2Physical Education and Sport Section, University of Lleida, Av. De l’Estudi General, 25001 Lleida, Spain

**Keywords:** public health, bibliometrics, productivity, network analysis, citation analysis, Web of Science

## Abstract

(1) Background: The main basis for the public recognition of the merits of scientists has always been the system of scientific publications and citations. Our goal is to identify and analyze the most cited articles in the Public, Environmental & Occupational Health category. (2) Methods: We searched the Web of Science for all articles published in the “Public, Environmental & Occupational Health” category up to March 2022 and selected the 100 most cited articles. We recorded the number of citations, the journal, the year of publication, quartile, impact factor, institution, country, authors, topic, type of publication and collaborations. (3) Results: 926,665 documents were analyzed. The top 100 had 401,620 citations. The journal with the most articles was the *Journal of Clinical Epidemiology* and the one with the highest number of citations was Medical Care. The year with the highest number of articles in the top 100 was 1998. The country with the highest percentage of publications was the USA and the most productive institution was Harvard. The most frequent keywords were bias, quality, and extension. The largest collaboration node was between the USA, Canada, Germany, Spain, Australia, France, and Sweden. (4) Conclusions: This bibliometric study on Public, Environmental & Occupational Health provides valuable information not only to identify topics of interest in the analyzed category, but also to identify the differences in the topics they study.

## 1. Introduction

Bibliometrics is a science that uses statistical and mathematical procedures to track the general trend of research in a specific field [1]. Various authors have targeted the participation of researchers in scientific activities, as well as differences and conditioning factors from the different fields of scientific knowledge [2]. These authors attribute different frequency and different publication practices between scientific disciplines [2]. The Web of Science (WoS) online database includes all important research papers and provides integrated analysis tools to produce representative figures, that is, it is the reference database of institutions, researchers and actors linked to science [3,4,5].

Within bibliometrics, citation analysis is one of the most used tools to assess the academic impact of an article in a specific area of knowledge [6]. The number of citations a publication receives does not necessarily reflect the quality of the research or the relevance of its authors [7], but it has been suggested that articles with the highest number of citations may have the ability to generate changes in practice, controversy, discussion and more research [6,8,9], or, as suggested by Zhu et al. [1], the number of citations can measure the article’s influence and merit. In addition, WoS search results could be exported to software for later analysis such as VOS-viewer [10], which could provide important information associated with collaboration networks between countries, institutions or authors. 

Although there have been bibliometric analyses of articles in the field of food safety [11,12]; environmental health [13,14]; health promotion [15,16]; health education [17]; mental health [18,19]; sport health [20]; and occupational health [21,22], the entire category of Public, Environmental & Occupational Health has never been studied worldwide. 

Few studies have a standardized measure of the wide range of dissemination activities in a scientific category that allows a detailed observation of production, collaboration and interrelation in a scientific field. No explorations have been performed based on quantitative methodologies aimed at building indicators on which to be able to empirically test the scientific productivity of the Public, Environmental & Occupational Health category. 

To our knowledge, there is no study that bibliometrically analyzes high citation articles that evaluate the Public, Environmental & Occupational Health category. Therefore, this study aimed to identify and analyze the 100 most cited articles in the Public, Environmental & Occupational Health category to understand the historical perspective and promote discussion and scientific progress in this specialty. 

## 2. Materials and Methods

### 2.1. Search Strategy and Eligible Criteria

Bibliometric analysis was performed on 14 March 2022. Two independent researchers, who searched the Web of Science Core Collection (Clarivate Analytics), a research platform that provides a substantial bibliographic database through of Science Citation Index Expanded (SCIE) using the search category, identified articles. The search strategy was performed through the “Public, Environmental & Occupational Health” category. We refined the research by selecting original research articles and reviews. The 100 articles with the most citations were eligible for bibliometric analysis, arranged in descending order of citation count. Any disagreement between the reviewers was discussed between them to reach a final decision. Author in descending order according to the number of citations ordered these articles.

### 2.2. Data Extraction

Two authors independently retrieved information from all articles. Through the Web of Science, the 100 articles with the highest number of citations were selected and exported. Later, they were exported into an Excel document where the following were recorded: the number of citations, name of the journal, year of publication, first and last author and co-authors, total number of authors, geographical location, origin and associated institute, the title of the article, type of document (article or review), abstract and corresponding author. For the analysis of authors, all the authors who participated in the study were counted. In the bibliometric analysis by country, each country that participated in the study was taken into account and the citations received were counted. Citations received by a country more than once were not counted if several authors from different institutions but from the same country had participated in the same study. The number of articles per country was counted as long as there was an author from the country in the study. If the first author was affiliated with two institutions, then the first institute was selected for inclusion.

### 2.3. Statical Analysis

We used IBM SPSS Statistics for Windows, Version 27.0 (Armonk, NY, USA: IBM Corp.) for correlation analysis. Correlation was determined using Pearson’s correlation coefficient (r), and when *p* < 0.05, the difference was considered statistically significant. We used a popular bibliometric analysis tool, VOSviewer 1.6.18 software (CWTS, Leiden, The Netherlands) [23], for cooperative network identification and keyword co-occurrence analysis. In addition, it could generate visual maps of knowledge. We also used the MapChart program [24], a platform from which a personalized map of different regions of the world was created, using colors and descriptions.

## 3. Results

The study flowchart is shown in Figure 1, and included studies that were published from 1900 to 2022 for the Public, Environmental & Occupational Health category of Web of Science. The search topic, after applying the strategy, produced 926,665 documents. For the analysis of the study, only articles or review articles were taken into consideration, which led to the exclusion of 294,361 documents. Of the remaining 632,304, the 100 documents with the highest number of citations were considered for the study. A total of 632,204 documents were excluded.

### 3.1. Publication Year, Citation and Bibliometric Analysis of the Keywords

The 100 most cited publications in the Public, Environmental & Occupational Health category were published between 1938 and 2020, of which 70% were published after 2018. We performed an analysis of publication trends by 6-year intervals based on a ranking of publication dates. Between 1998 and 2003, 29 documents were published, with the year 1998 (*n* = 11) being the year of greatest production. There has been a visible improvement in the quantity of the data, since, of the 100 articles, before 1998 a total of 29 documents were published, making a big difference with the period 1998 to 2003, when 29 articles were published (Figure 2).

The top 100 articles were cited 401,620 times in total, and the average total number of citations was 4016 citations (ranging from 1846 to 30,229). No significant correlation was found between the total number of citations and the age of the articles (r = −0.121, *p* = 0.229). The most cited article (30,229 citations) was “A new method of classifying prognostic co-morbidity in longitudinal-studies-development and validation” by Charlson et al. [25] published in the *Journal of Chronic Diseases*. Based on the number of publications in the 100 articles, and analyzing the citations per publication, 1998 was the most productive year with 11 articles (42,320 citations and an average of 3847 citations/article) in the top 100 list (Table 1).

The oldest study included in the list was published by Miles et al. [26] in 1938 entitled “The estimation of the bactericidal power of the blood”, with 3246 citations. The last study included was published in 2020 by Wang et al. [27], the paper entitled “Immediate Psychological Responses and Associated Factors during the Initial Stage of the 2019 Coronavirus Disease (COVID-19) Epidemic among the General Population in China”, with 2485 citations, published in the *International Journal of Environmental Research and Public Health* (Table 1).

Eighty-six of the 100 publications were original research, and the remaining 14 were reviews. The average number of citations per article in the review works was 3285 citations/article compared to 4135 citations/article in the original works (Table 1). The most common important keywords included quality of life, comorbidity, disease, cancer, clinical-trials, bias and epidemiology, and the keywords that appeared the most were “bias” (total link strength of 14), “quality” (total link strength of 11) and “extension” (total link strength of 10), which had a strong link with “epidemiology”, “metaanalysis” and “cancer” (Figure 3).

### 3.2. Authors and Bibliometric Analysis of the Co-Authorship

A total of 487 authors contributed the 100 most cited. The number of authors in an article ranged from 1 to 26 (mean 5.53). Analysis of the 10 most productive authors based on their number of articles in the top 100, regardless of their authorship positions, showed that Ware, J.E., Altman, D.G. and Horan, T.C. were the authors with the highest number of articles.

Ware, J.E., from the USA, had a maximum of 46,062 citations with five articles listed and an h-index of 100. The average number of citations/article was 9212 citations. However, Altman, D.G. from England, published four papers, the total index of citations was 36,420 and the average per article was 9105; their h-index was 182. The third position is for the researcher Horan, T.C. from the USA, with four published documents, an h-index of 25, and with more than 13,500 total citations (Table 2).

The total number of citations was not related to the number of authors (r = −0.118, *p* = 0.058). However, the average number of citations per article was associated with the number of authors (r = 0.210, *p* < 0.001).

There was low collaboration between most of the main authors, creating only one cooperation research network. Authors with a minimum of three papers per author were considered for analysis. Of the 487 authors, seven reached the threshold (Figure 4). Altman, D.G. formed a collaborative network with five other researchers with a link strength of 15. 

### 3.3. Countries, Institutions and Bibliometric Analysis of the Collaboration

A total of 26 countries published the 100 most cited articles in the Public, Environmental & Occupational Health category. Table 3 shows the twelve most productive countries, with the USA being the one that contributes the most, with 65 documents, followed by England with 21 and Canada with 17 articles. These three same countries obtain also the greatest number of citations. However, the country with the highest rate of citations per article is Italy, with an average of 5151 citations/article, followed by England with an average of 4724 citations/article.

Two collaboration nodes were established. A larger one, involving seven countries and where the USA had the most active partnership (a liaison force of 45 and collaborated on 57 documents); its major research cooperators included Canada, Germany, Spain, Australia, France, Sweden. The other node was where England (with a link strength of 37 and 20 documents) had a strong collaboration with mainly European countries such as Denmark, The Netherlands and Switzerland. We found that Italy and Norway rarely cooperated with other countries in investigations. (Figure 5).

The world map revealed that the articles were mainly concentrated in North American and western Europe, and less so in Oceania. Specifically, the USA was the country with the highest production of documents, followed by England and Canada (Figure 6).

In total, 228 institutions participated in the 100 articles. The number of institutions per article ranged between 1 and 21. The average institutional collaboration was 3.8 institutions/article. The article with 21 participating institutions was a review on toxic equivalency factors (TEFs) published in 1998 with 2592 citations. The World Health Organization, with eight articles included in this bibliometric analysis, was the institution with the greatest scientific representation (Table 4). In four of the eight papers, it was included as the main institution of the study. The total number of citations was 20,339 and the average number of citations per article was 2542 citations. The second institution was the University of Harvard in the USA with six documents and one as the main institution. The total number of citations was more than 25,000 with an average of 4209 citations per article.

There was a strong and significant correlation between the number of institutions and authors (r = 0.848, *p* < 0.001). There was a negative correlation between the total citations and the number of participating institutions (r = −0.115, *p* = 0.286).

In the collaboration network analysis (Figure 7), a minimum of three collaborations between institutions were established, 19 reached the threshold and three cooperation network nodes were formed. In the first of them, McMaster University cooperated with institutions such as Harvard University, University of Washington and the University of Toronto and collaborated on five articles. The University of Washington had a strong partnership and cooperation with the University of Minnesota, NCI, and Wake Forest University. Collaboration network analysis also highlighted the institutional collaboration network that the World Health Organization has with University of Toronto, Wisconsin University and US EPA, with more than seven documents shared. 

### 3.4. Journal Analysis

The Web of Science “Public, Environmental & Occupational Health” category had 204 indexed journals, of which 35 made it to the top 100 most cited articles list. A total of 16 journals were in the first quartile (approximately 45%), 9 journals in the second quartile, 3 in the third quartile, and 5 were from Q4. Two journals were out of print or had changed their name. A total of 79% of the studies were published in high-impact journals (Q1 or Q2).

The IFs of the 35 journals ranged from 0.875, *Malawi Medical Journal*, to 59.769, *MMWR Surveillance Summaries*. There were up to 22 journals with an IF < 5.000, seven between 5.000 and 10.000, and four journals with an IF > 10.000.

Table 5 shows the top nine journals that published three or more articles.

*Journal of Clinical Epidemiology* was the most productive journal (*n* = 15), followed by *Medical Care and Statistics in Medicine* (*n* = 12). The top five journals published 54% of the articles and account for more than 59% of the total citations. The self-citation rate for the top nine journals ranged from 1.7% for the *Bulletin of the World Health Organization* to 10.3% for the *Journal of Clinical Epidemiology*. The journal with the highest number of citations was *Medical Care* (*n* = 74,189) and its mean number of citations per article was 6182 citations/article.

## 4. Discussion

This is the first paper that analyzes the 100 most cited papers in the Public, Environmental & Occupational Health category of Web of Science. This article identifies the authors, journals, countries, institutions, etc., with the greatest impact in this category from the beginning of the 20th century to the present. The sample size was set at 100 manuscripts to provide a manageable and significant number of articles to be analyzed, in accordance with several published works [1,6,8,9,28,29].

The period of the greatest publication of articles starts in 1998; a clear upward trend in the production of works started during the period 1989–2010, but then it disappears in the last decade. A stochastic process is observed. The Mann–Kendall trend test (Figure 3) revealed a significant positive trend towards a greater number of articles over the years starting in 1985 (*p* = 0.055, Kendell’s Tauβ). Our results would be in line with those found by the authors of [6,29,30]. They contrasts with recent reviews, on other topics and specialties, in which most of the most cited papers were published earlier in the 1980s [31] or later, from the year 2000 [32,33]. The socioeconomic growth of recent years may be one of the causes of the advancement in scientific research, an evolution that the dissemination and communication of science has already been experiencing as an exponential change for some time. To understand these changes, it is necessary to know how science spreads. In the professional field, one of the main ways that the research community has to disseminate its work is the publication of scientific articles; however, on the other hand, they also use social networks and all Internet options (scientific forums, blogs, etc.). These tools are also protagonists in recent years, which encourage more dissemination of science and, therefore, more knowledge of what is published [34]. Some experts believe that studies that are more recent are cited today due to the advancement in scientific dissemination [10]. It may seem surprising that the studies with the highest number of citations are recent studies; among other factors, this could be due to the appearance of scientific journals in an electronic format, facilitating access and thus favoring circulation in the scientific community [6].

Some specialists consider that research goes further, suggesting the publication of an article ends when it is read and understood by a large part of society, that is, it is not enough just to publish, it is necessary for the audience to clearly understand its content and, thus, be able to cite it [35].

The keyword co-occurrence analysis found that the words “cancer”, “quality of life”, “comorbidity”, “epidemilogy” and “disease” had the highest frequency of co-occurrence in the research in the analyzed category. Our work reflects, in part, a growing trend in public health research. Studies on quality of life, comorbidity or cancer have been the focus of research in the scientific community and specifically within the Public, Environmental & Occupational Health category [36,37]. Performing a quick bibliographic search in WoS, these terms occupy the fourth, eighth and fourteenth position, respectively, with more records among the different categories.

Metadata from all documents were used to reveal the most productive authors and the most impactful sources. The high number of authors (487) contributing to the 100 articles, with more than an average of five authors per article, made it difficult to determine the individual contribution and, consequently, the role of each author [38]. As suggested by Bruni et al. [33], traditionally, in multi-author articles, the first position is occupied by the main contributor, while the last position is reserved for the supervisor. The authors with most impact in the studied category generally held relevant positions, either as main author or as supervisor. This is becoming more common due to the influence of experimental sciences, considering the same importance to the first and last author, based on the author/director relationship. This interpretation is known as the FLAE approach, an expression of first last author emphasis [39].

The h-index quantifies the research performance of individual scientists, incorporating both the number and visibility of publications [40]. In the work, we can see an unequal distribution of the h index among the authors of the 100 articles, where the number of citations that a scientific subcommunity grants to a manuscript is undoubtedly and directly related to the number of researchers that make up such a sub-community [41]. The analyzed category is a very broad field of knowledge; therefore, the number of citations of the articles will be very different depending on the topic analyzed.

As indicated by Jung et al. [42], in a context in which there is great interest in intensifying international collaboration within scientific practice, this paper proposes an approach on how to measure and visualize international collaborative work at the institutional level. The low collaboration observed between the different authors in our work contrasts with that found in studies such as by Zhu et al. [1] or Yu et al. [10]. The joint analysis of the collaboration indexes of the relationships between the different authors of the documents allows us to make a better interpretation of the structure of international scientific collaboration networks in the category of study [43]. One of the variables handled in our work was the possibility of identifying whether there was a high level of international and potentially multinational collaboration with other institutions that could affect the visibility of the research and the frequency of citations of a category [44]. This was not the case, but we have been able to map and identify the existing collaborations within the Public, Environmental & Occupational Health field, as well as the main citation sources.

The most relevant works were mainly in North America, specifically the USA and Canada, and Western Europe. Similarly, citation analysis showed this same trend in previous studies [45,46]. This trend can be explained by several reasons, first of all, by the cumulative geographical advantage, since citations originate more frequently from institutions located in the same country as the place of residence of the author [47,48]. Second, as suggested by Wang et al. [49], the USA can count on a broad scientific community and generous science funding policies. In fact, the most productive institutions in our study are geographically located in the USA and Europe. A third reason may be that larger universities provide greater opportunities for scientists to collaborate and work on similar topics, and co-authorship may lead to higher citation rates.

Most items originated from two major advanced economies: North America and Western Europe. These are undoubtedly economically developed continents with more access to early research and they can support medical research [29]. This can be seen in the data published by WHO in a report published in 2020, in which high-income countries spend a higher percentage of GDP on R&D in the health sector [50].

The cooperative network of research institutions can reveal the distribution of research forces in the field of Public, Environmental & Occupational Health. The USA has the most extensive cooperative relationships and prefers to cooperate with Canada and some European countries. England, with the second highest number of co-authored articles, prefers to work with other European countries. Our results would be in line with those found by Song et al. [51] in the Entrepreneurship research area. In the field of science, collaborative work, institutional and disciplinary structures face the challenge of a global context. This challenge has led to the creation of initiatives such as e-Science in the United Kingdom, which was announced as a global collaboration program in key areas of science, and the development of the next generation of infrastructure. These types of initiatives show that contemporary scientific practice is characterized by being very collaborative, multidisciplinary, global work with intensive data management [52].

According to the results, more than half of the classified articles were published in only five journals, collecting more than half of the total citations. These results demonstrate that a significant number of studies concentrated on a limited core of journals, in accordance with Bradford’s law [53]. As indicated by Highhouse et al. [54], authors tend to send their work to the most prestigious journals, attracted, according to Bruni et al. [33], due to the greater visibility in the search results, as well as the greater probability of being cited.

If we look at the quartiles of the journals, works mostly appears on first and second quartile journals. As stated by Torres-Salinas and Cabezas-Calvijo [55], publication in high-impact journals generates benefits, starting with the fact that a scientist who regularly publishes in these journals will be able to advance smoothly in his scientific career and will be recognized as an expert in his field. Other authors affirm that publication in high-impact journals helps to develop one’s own criteria, increases self-esteem, strengthens the confidence of the researcher, and feeds the desire to continue researching and publishing, in addition to guaranteeing quality through arbitration, such as peer review demonstrates [35,56].

There is no doubt that the use of the JIF as an evaluation measure generates debate, but today it is a useful way to measure the prestige and importance of scientific journals in the international system, as well as for their researchers [57]. Many authors have pointed out that the JIF has some limitations such as: (a) a built-in bias that favors American journals (in the case of our study, six of the nine journals in the ranking are published in the USA), (b) scoring highly variable IF between fields and specialties within fields, (c) vulnerability to inflation due to self-citation of journals, (d) vulnerability to inflation due to the publication of review articles and meta-analyses, and, finally, (e) an arbitrary citation window that penalizes some fields or specialties within them [54,57].

Finally, we wanted to also compare the JIF without the self-citations. In this sense, the level of self-citation of the analyzed journals was relatively low, with some exceptions. The abuse of self-citations is another element that can substantially affect the JIF. The self-citation rate in the presented list was low (8.1%) compared with other studies [33,58,59]. This is a bias that many platforms have been working on for years to solve [60,61].

## 5. Conclusions

The work allows the identification of relevant aspects in order to encourage scientific mapping in the Public, Environmental & Occupational Health category. The analysis can help the governance of specific areas or it can outline an institution’s research. The category analyzed has very varied topics; however, it allowed us to identify the most cited authors, institutions with greater visibility, and the most notable articles.

It has also made it possible to analyze the researchers who are forming national and international collaboration networks, as well as to identify the most collaborative authors and institutions.

Currently, the publication rate of American researchers is the highest in the category studied and its institutions are among the most productive. In addition, the collaborative network of countries, institutions and authors shows the influence of European and American countries in the Public, Environmental & Occupational Health category.

Keyword analysis was an effective method to identify interesting topics among researchers and mark research trend lines.

The results of this research open up new possibilities to identify new strategies and institutional policies that allow them to consolidate their research networks.

Although there has been an exponential growth in work, greater efforts are still required from both researchers and institutions.

In this article, valuable information is provided not only to identify topics of interest in the analyzed category, but also to identify the differences on topics studied between the areas that form the category.

## Figures and Tables

**Figure 1 ijerph-19-09645-f001:**
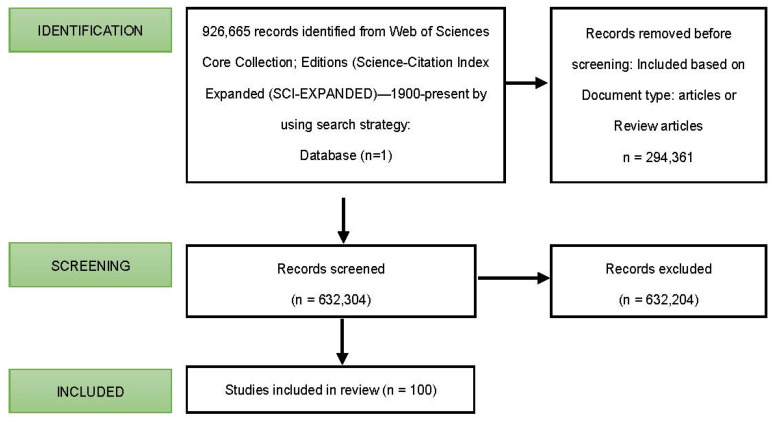
Flowchart of study.

**Figure 2 ijerph-19-09645-f002:**
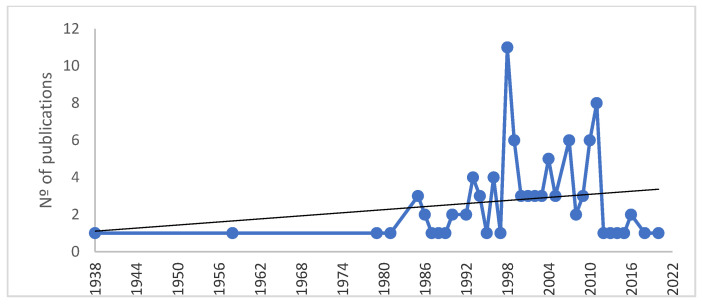
Pattern of distribution of top-cited articles (number of articles per year).

**Figure 3 ijerph-19-09645-f003:**
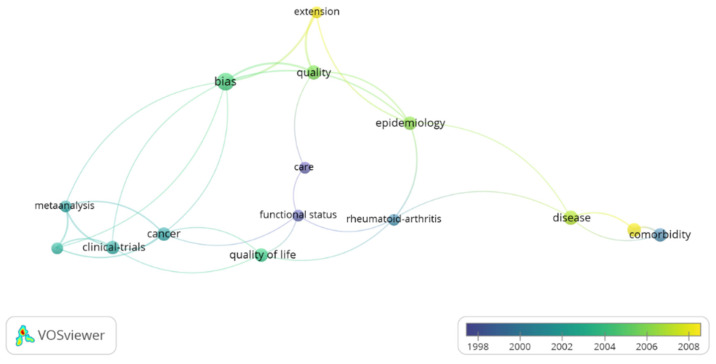
The co-occurrence network of keywords. **Note**: The size of the nodes indicates the frequency of occurrence. The curves between the nodes represent their co-occurrence in the same publication. The smaller the distance between two nodes, the higher the number of co-occurrence of the two keywords.

**Figure 4 ijerph-19-09645-f004:**
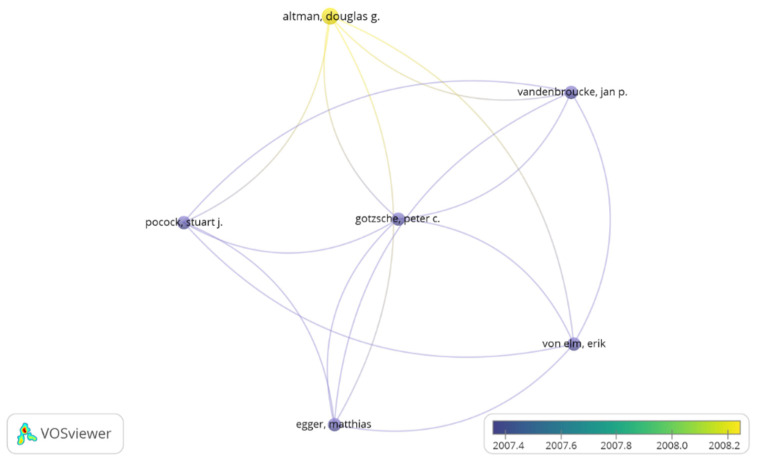
The author collaboration network. **Note:** The collaboration map of authors reflects the scientific research cooperation between them. The circle/node signifies the authors; size of the circle/node signifies the number of articles. The lines denote the authors’ collaboration strength, and each color signifies a cluster.

**Figure 5 ijerph-19-09645-f005:**
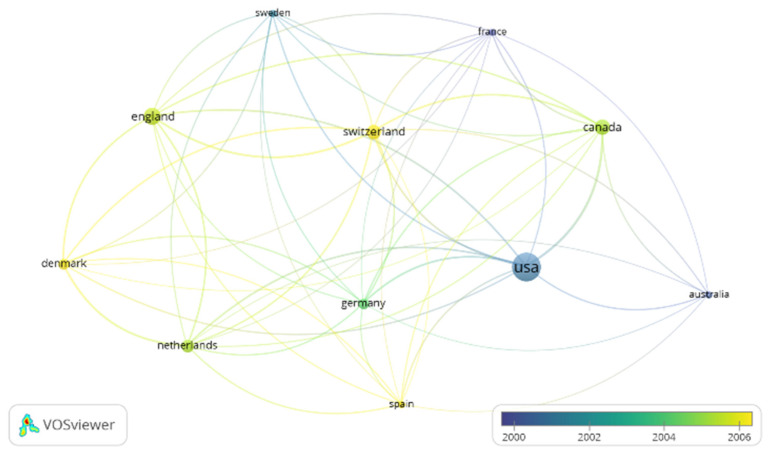
The country collaboration network.

**Figure 6 ijerph-19-09645-f006:**
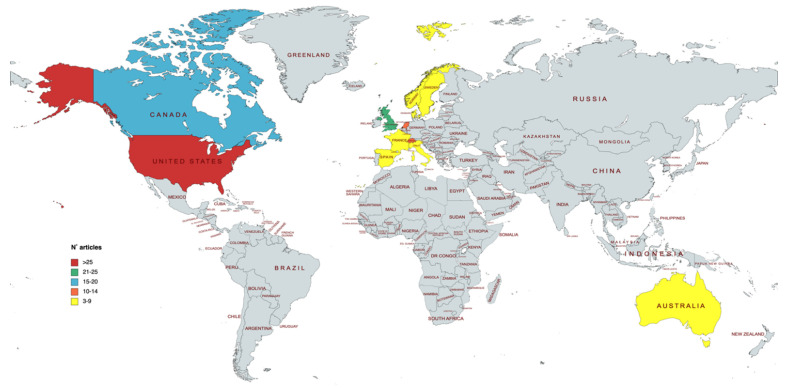
The distribution map of the number of published articles worldwide for countries (MapChart).

**Figure 7 ijerph-19-09645-f007:**
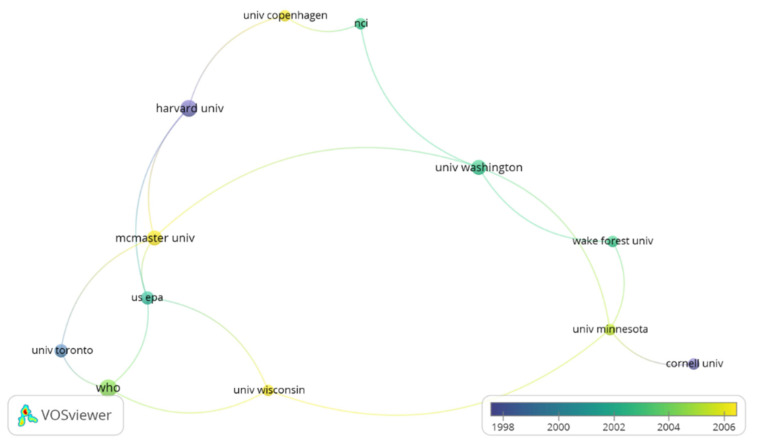
The institution collaboration network.

**Table 1 ijerph-19-09645-t001:** The top 100 articles with most total citations in Public, Environmental & Ocupational Health category.

Ranking Position	Times Cited, WoS Core	First Author	Article Title	Source Title	Country	Publication Year
1	30,229	Charlson, ME	A new method of classifying prognostic co-morbidity in longitudinal-studies-development and validation	*Journal of Chronic Diseases*	USA	1987
2	24,802	Ware, JE	The Mos 36-Item short-form health survey (SF-36). 1. Conceptual-framework and item selection	*Medical Care*	USA	1992
3	18,779	Higgins, JPT	Quantifying heterogeneity in a meta-analysis	*Statistics in Medicine*	England	2002
4	11,110	Ware, JE	A 12-item short-form health survey—Construction of scales and preliminary tests of reliability and validity	*Medical Care*	USA	1996
5	11,091	von Elm, E	The Strengthening the Reporting of Observational Studies in Epidemiology (STROBE) statement: Guidelines for reporting observational studies	*Preventive Medicine*	Switzerland	2007
6	10,754	Schulz, KF	CONSORT 2010 Statement: Updated guidelines for reporting parallel group randomised trials	*Journal of Clinical Epidemiology*	USA	2010
7	10,302	Moher, D	Preferred Reporting Items for Systematic Reviews and Meta-Analyses: The PRISMA Statement	*Journal of Clinical Epidemiology*	Canada	2009
8	7879	Deyo, RA	Adapting a clinical comorbidity index for use eith ICD-9-CM administrative databases	*Journal of Clinical Epidemiology*	USA	1992
9	7582	Felitti, VJ	Relationship of childhood abuse and household dysfunction to many of the leading causes of death in adults—The adverse childhood experiences (ACE) study	*American Journal of Preventive Medicine*	USA	1998
10	6438	Harrell, FE	Multivariable prognostic models: Issues in developing models, evaluating assumptions and adequacy, and measuring and reducing errors	*Statistics in Medicine*	USA	1996
11	6116	Elixhauser, A	Comorbidity measures for use with administrative data	*Medical Care*	USA	1998
12	5658	Quan, HD	Coding algorithms for defining comorbidities in ICD-9-CM and ICD-10 administrative data	*Medical Care*	Canada	2005
13	5138	Oberdorster, G	Nanotoxicology: An emerging discipline evolving from studies of ultrafine particles	*Environmental Health Perspectives*	USA	2005
14	5111	Terwee, CB	Quality criteria were proposed for measurement properties of health status questionnaires	*Journal of Clinical Epidemiology*	Netherland	2007
15	5034	Zou, GY	A modified Poisson regression approach to prospective studies with binary data	*American Journal of Epidemiology*	Canada	2004
16	4838	McHorney, CA	The Mos 36-Item Short-form Health survey (SF-36). 2. Psychometric and clinical-test of validity in measuring physical and mental-health constructs	*Medical Care*	USA	1993
17	4707	Downs, SH	The feasibility of creating a checklist for the assessment of the methodological quality both of randomised and non-randomised studies of health care interventions	*Journal of Epidemiology and Community Health*	England	1998
18	4636	Garner, JS	CDC Definitions for nosocomial infections, 1988	*American Journal of Infection Control*	USA	1988
19	4451	Pencina, MJ	Evaluating the added predictive ability of a new marker: From area under the ROC curve to reclassification and beyond	*Statistics in Medicine*	USA	2008
20	4356	Peduzzi, P	A simulation study of the number of events per variable in logistic regression analysis	*Journal of Clinical Epidemiology*	USA	1996
21	4273	von Elm, E	The Strengthening the Reporting of Observational Studies in Epidemiology (STROBE) statement: guidelines for reporting observational studies	*Journal of Clinical Epidemiology*	Switzerland	2008
22	4143	White, IR	Multiple imputation using chained equations: Issues and guidance for practice	*Statistics in Medicine*	England	2011
23	4068	D’Agostino, RB	Propensity score methods for bias reduction in the comparison of a treatment to a non-randomized control group	*Statistics in Medicine*	USA	1998
24	3962	Charlson, M	Validation of a combined comorbidity index	*Journal of Clinical Epidemiology*	USA	1994
25	3947	Horan, TC	CDC/NHSN surveillance definition of health care-associated infection and criteria for specific types of infections in the acute care setting	*American Journal of Infection Control*	USA	2008
26	3845	Guyatt, GH	GRADE guidelines: 9. Rating up the quality of evidence	*Journal of Clinical Epidemiology*	Canada	2011
27	3816	Ahlbom, A	Guidelines for limiting exposure to time-varying electric, magnetic, and electromagnetic fields (up to 300 GHz)	*Health Physics*	Germany	1998
28	3783	Caspersen, CJ	Physical-activity, exercise, and physical-fitness—definitions and distrinctions for health-related research	*Public Health Reports*	USA	1985
29	3755	de Onis, M	Development of a WHO growth reference for school-aged children and adolescents	*Bulletin of the World Health Organization*	Switzerland	2007
30	3495	Balshem, H	GRADE guidelines: 3. Rating the quality of evidence	*Journal of Clinical Epidemiology*	USA	2011
31	3385	McHorney, CA	The Mos 36-Item short-form health survey (SF-36). 3. Tests of data quality, scaling assumptions, and reliability across diverse patient groups	*Medical Care*	USA	1994
32	3380	Willett, WC	Reproducibility and validatity of a semiquantitative food requency questionnaire	*American Journal of Epidemiology*	USA	1985
33	3363	Guillemin, F	Cross-cultural adaptation of health-related quality-of-life mesures—literatures—review and proposed guidelines	*Journal of Clinical Epidemiology*	Canada	1993
34	3361	Bergner, M	The sickness impact profile—development and final revision of a health-status measure	*Medical Care*	USA	1981
35	3246	Miles, AA	The estimation of the bactericidal power of the blood	*Journal of Hygiene*	Canada	1938
36	3223	Parmar, MKB	Extracting summary statistics to perform meta-analyses of the published literature for survival endpoints	*Statistics in Medicine*	Italy	1998
37	3198	Daughton, CG	Pharmaceuticals and personal care products in the environment: Agents of subtle change?	*Environmental Health Perspectives*	USA	1999
38	3174	Herdman, M	Development and preliminary testing of the new five-level version of EQ-5D (EQ-5D-5L)	*Quality of Life Research*	Spain	2011
39	3162	Dolan, P	Modeling valuations for EuroQol health states	*Medical Care*	England	1997
40	3155	Hudak, PL	Development of an upper extremity outcome measure: The DASH (Disabilities of the Arm, Shoulder, and Head)	*American Journal of Industrial Medicine*	Canada	1996
41	3133	Berkman, LF	Social networks, host-resistance, and mortality—9- year follou-up-study of alameda county residents	*American Journal of Epidemiology*	USA	1979
42	3079	Morisky, DE	Concurrent and predictive-validity of a self-reported measure of medication adherence	*Medical Care*	USA	1986
43	3039	Clarke, DH	Techniques for hemagglutination and hemagglutination-inhibition with arthropod-borne viruses	*American Journal of Tropical Medicine and Hygiene*	Ireland	1958
44	3028	Israel, BA	Review of community-based research: Assessing partnership approaches to improve public health	*Annual Review of Public Health*	USA	1998
45	3007	Robins, JM	Marginal structural models and causal inference in epidemiology	*Epidemiology*	USA	2000
46	2976	Andresen, EM	Screening for depression in well older adults—evaluation of a short-form of the CES-D	*American Journal of Preventive Medicine*	USA	1994
47	2946	Varni, JW	PedsQL (TM) 4.0: Reliability and validity of the pediatric quality of life Inventory (TM) Version 4.0 generic core scales in healthy and patient populations	*Medical Care*	USA	2001
48	2917	Mangram, AJ	Guideline for Prevention of Surgical Site Infection, 1999	*Infection Control and Hospital Epidemiology*	USA	1999
49	2883	Kroenke, K	The Patient Health Questionnaire-2—Validity of a two-item depression screener	*Medical Care*	USA	2003
50	2878	Glasgow, RE	Evaluating the public health impact of health promotion interventions: The RE-AIM framework	*American Journal of Public Health*	USA	1999
51	2849	Norman, GR	Interpretation of changes in health-related quality of life—The remarkable universality of half a standard deviation	*Medical Care*	Canada	2003
52	2848	Newcombe, RG	Interval estimation for the difference between independent proportions: Comparison of eleven methods	*Statistics in Medicine*	Wales	1998
53	2743	Ludvigsson, JF	External review and validation of the Swedish national inpatient register	*BMC Public Health*	Sweden	2011
54	2714	Kim, HJ	Permutation tests for joinpoint regression with applications to cancer rates	*Statistics in Medicine*	USA	2000
55	2714	Colborn, T	Delepmental effects of endocrine-disrupting chemicals in wildlife and humans	*Environmental Health Perspectives*	USA	1993
56	2688	Resnikoff, S	Global data on visual impairment in the year 2002	*Bulletin of the World Health Organitzation*	Switzerland	2004
57	2592	Van den Berg, M	Toxic equivalency factors (TEFs) for PCBs, PCDDs, PCDFs for humans and wildlife	*Environmental Health Perspectives*	Netherland	1998
58	2589	Lynge, E	The Danish National Patient Register	*Scandinavian Journal of Public Health*	Denmark	2011
59	2574	Williams, OD	The atherosclerosis risk in communities (ARIC) study—Deseign and objectives	*American Journal of Epidemiology*	USA	1989
60	2517	Willett, W	Total energy-intake—implications for epidemiologic analyses	*American Journal of Epidemiology*	USA	1986
61	2485	Wang, CY	Immediate Psychological Responses and Associated Factors during the Initial Stage of the 2019 Coronavirus Disease (COVID-19) Epidemic among the General Population in China	*International Journal of Environmental Research and Public Health*	China	2020
62	2470	FerroLuzzi, A	Physical status: The use and interpretation of anthropometry—Introduction	*Physical Status: The use and Interpretation of Anthropometry*	USA	1995
63	2460	Austin, PC	Balance diagnostics for comparing the distribution of baseline covariates between treatment groups in propensity-score matched samples	*Statistics in Medicine*	Canada	2009
64	2452	Pedersen, CB	The Danish Civil Registration System	*Scandinavian Journal of Public Health*	Denmark	2011
65	2435	Cai, ZJ	WHO expert committee on drug dependence—Thirty-first report—Introduction	*WHO Expert Committe on Drug Dependec—31 St Report*	China	1999
66	2413	Baumgartner, RN	Epidemiology of sarcopenia among the elderly in New Mexico	*American Journal of Epidemiology*	USA	1998
67	2356	Quan, HD	Updating and Validating the Charlson Comorbidity Index and Score for Risk Adjustment in Hospital Discharge Abstracts Using Data From 6 Countries	*American Journal of Epidemiology*	Canada	2011
68	2304	Bild, DE	Multi-ethnic study of atherosclerosis: Objectives and design	*American Journal of Epidemiology*	USA	2002
69	2271	Steyerberg, EW	Assessing the Performance of Prediction Models A Framework for Traditional and Novel Measures	*Epidemiology*	Netherland	2010
70	2270	Mukaka, MM	Statistics Corner: A guide to appropriate use of Correlation coefficient in medical research	*Malawi Medical Journal*	England	2012
71	2255	Workowski, KA	Sexually Transmitted Diseases Treatment Guidelines, 2015	*MMWR Recommendations and Reports*	USA	2015
72	2255	Peppard, PE	Increased Prevalence of Sleep-Disordered Breathing in Adults	*American Journal of Epidemiology*	USA	2013
73	2220	Skevington, SM	The World Health Organization’s WHOQOL-BREF quality of life assessment: Psychometric properties and results of the international field trial—A report from the WHOQOL group	*Quality of Life Research*	England	2004
74	2181	Woolf, AD	Burden of major musculoskeletal conditions	*Bulletin of the World Health Organitzation*	England	2003
75	2154	Cohen, SH	Clinical Practice Guidelines for Clostridium difficile Infection in Adults: 2010 Update by the Society for Healthcare Epidemiology of America (SHEA) and the Infectious Diseases Society of America (IDSA)	*Infection Control and Hospital Epidemiology*	USA	2010
76	2143	Gooley, TA	Estimation of failure probabilities in the presence of competing risks: New representations of old estimators	*Statistics in Medicine*	USA	1999
77	2135	Cella, D	The Patient-Reported Outcomes Measurement Information System (PROMIS) developed and tested its first wave of adult self-reported health outcome item banks: 2005–2008	*Journal of Clinical Epidemiology*	USA	2010
78	2132	Greenland, S	Causal diagrams for epidemiologic research	*Epidemiology*	USA	1999
79	2125	Klepeis, NE	The National Human Activity Pattern Survey (NHAPS): a resource for assessing exposure to environmental pollutants	*Journal of Exposure Analysis and Environmental Epidemiology*	USA	2001
80	2093	Smith, GD	‘Mendelian randomization’: can genetic epidemiology contribute to understanding environmental determinants of disease?	*International Journal of Epidemiology*	England	2003
81	2082	Cardo, D	National Nosocomial Infections Surveillance (NNIS) System Report, data summary from January 1992 through June 2004, issued October 2004	*American Journal of Infection Control*	USA	2004
82	2010	Dowell, D	CDC Guideline for Prescribing Opioids for Chronic Pain—United States, 2016	*MMWR Recommendations and Reports*	USA	2016
83	1998	Rose, G	Sick individuals and sick populations	*International Journal of Epidemiology*	England	1985
84	1997	Vittinghoff, E	Relaxing the rule of ten events per variable in logistic and Cox regression	*American Journal of Epidemiology*	USA	2007
85	1980	Washburn, RA	The physical-activity scale for the elderly (PASE)—Development and evaluation	*Journal of Clinical Epidemiology*	USA	1993
86	1969	Guh, DP	The incidence of co-morbidities related to obesity and overweight: A systematic review and meta-analysis	*BMC Public Health*	Canada	2009
87	1968	Baio, J	Prevalence of Autism Spectrum Disorder Among Children Aged 8 Years—Autism and Developmental Disabilities Monitoring Network, 11 Sites, United States, 2014	*MMWR Surveillance Summaries*	USA	2018
88	1960	Torre, LA	Global Cancer Incidence and Mortality Rates and Trends-An Update	*Cancer Epidemiology Biomarkers & Prevention*	USA	2016
89	1932	Sterne, JAC	Funnel plots for detecting bias in meta-analysis: Guidelines on choice of axis	*Journal of Clinical Epidemiology*	England	2001
90	1927	Gandek, B	Cross-validation of item selection and scoring for the SF-12 Health Survey in nine countries: Results from the IQOLA Project	*Journal of Clinical Epidemiology*	USA	1998
91	1909	Thompson, SG	How should meta-regression analyses be undertaken and interpreted?	*Statistics in Medicine*	England	2002
92	1906	Slovic, P	Risk as analysis and risk as feelings: Some thoughts about affect, reason, risk, and rationality	*Risk Analysis*	USA	2004
93	1885	Wang, Y	The obesity epidemic in the United States—Gender, age, socioeconomic, Racial/Ethnic, and geographic characteristics: A systematic review and meta-regression analysis	*Epidemiology Review*	USA	2007
94	1876	Say, L	Global causes of maternal death: a WHO systematic analysis	*Lancet Global Health*	Switzerland	2014
95	1871	Rice, D	Critical periods of vulnerability for the developing nervous system: Evidence from humans and animal models	*Environmental Health Perspectives*	USA	2000
96	1868	Mokkink, LB	The COSMIN checklist for assessing the methodological quality of studies on measurement properties of health status measurement instruments: an international Delphi study	*Quality of Life Research*	Netherland	2010
97	1864	Reitsma, JB	Bivariate analysis of sensitivity and specificity produces informative summary measures in diagnostic reviews	*Journal of Clinical Epidemiology*	Netherland	2005
98	1856	Jemal, A;	Global Patterns of Cancer Incidence and Mortality Rates and Trends	*Cancer Epidemiology Biomarkers & Prevention*	USA	2010
99	1848	Feinstein, AR	High agreement but low Kappa. 1. The problem of 2 paradoxes	*Journal of Clinical Epidemiology*	USA	1990
100	1846	Hochberg, Y	More powerful procedures for multiple significance testing	*Statistics in Medicine*	Israel	1990

**Table 2 ijerph-19-09645-t002:** The top authors with the most articles in the top 100.

Authors	Number of Articles	H-Index	First Author	Last Author	Co-Author	Total Citations	Mean Citation per Article
Ware, JE	5	100	2		3	46,062	9212
Altman, DG	4	182		1	3	36,420	9105
Horan, TC	4	25	1		3	13,582	3396
Egger, M	3	30		1	2	17,296	5765
Charlson, M	2	58	2			34,191	17,096
Sherbourne, CD	2	66		2		28,187	14,094
Moher, D	2	21	1	1		21,056	10,528
Higgins, JPT	2	102	1	1		20,688	10,344
Thompson, SG	2	58	1	1		20,688	10,344
Gotzsche, PC	2	82			2	15,364	7682

**Table 3 ijerph-19-09645-t003:** The top countries with the most highly cited articles.

Addresses	Times Cited, WoS Core	Number Articles	Mean Citations per Article
USA	266,604	65	4102
England	99,202	21	4724
Canada	66,702	17	3924
Switzerland	44,620	12	3718
Netherlands	36,641	10	3664
Denmark	30,369	8	3796
Italy	15,452	3	5151
Australia	12,051	4	3013
Spain	10,814	4	2704
France	10,081	4	2520
Sweden	9697	4	2424
Norway	9267	3	3089

**Table 4 ijerph-19-09645-t004:** The top institutions with the most highly cited articles.

Institution	Country	Number Articles	Number of the First Institution	Total Citation	Mean Citation per Article
World Health Organization (WHO)	Switzerland & Netherlands	8	4	20,339	2542.4
Harvard University	USA	6	1	25,255	4209.2
University of Washington	USA	6	1	19,690	3281.7
McMaster University	Canada	5	2	15,212	3042.4
University of Columbia	USA	3	0	9904	3301.3
Center for Disease Control & Prevention	USA	4	1	20,264	5066.0
Johns Hopkins Bloomberg	USA	4	1	9232	2308.0
Tufts University	USA	4	2	18,055	4513.8
University of Bristol	England	4	2	19,389	4847.3
University of London	England	4	2	22,069	5517.3
Oxford University	England	4	0	36,420	9105.0
University of Toronto	Canada	4	0	11,413	2853.3
U.S. Environmental Protection Agency	USA	4	2	11,506	2876.5

**Table 5 ijerph-19-09645-t005:** The top journals that published the top 100 highly cited literature in Public, Environmental & Occupacional Health category.

Source Title	Records	Number Total Citation	% Total De Citation	Nnumber Citation for Paper	Impact Factor (2020)	IF without Self Citations	Quartile
** *Journal of Clinical Epidemiology* **	15	64,753	16.12	4317	6.437	5.771	Q1
** *Medical Care* **	12	74,189	18.47	6182	2.983	2.891	Q2
** *Statistics in Medicine* **	12	55,022	13.70	4585	2.373	2.149	Q3
** *American Journal of Epidemiology* **	10	27,963	6.96	2796	4.897	4.722	Q1
** *Environmental Health Perspectives* **	5	15,513	3.86	3103	9.031	8.657	Q1
** *Bulletin of the World Health Organization* **	4	12,897	3.21	3224	9.408	9.252	Q1
** *American Journal of Infection Control* **	3	10,665	2.66	3555	2.918	2.655	Q2
** *Epidemiology* **	3	7410	1.85	2470	4.822	4.623	Q1
** *Quality of Life Research* **	3	7262	1.81	2421	4.147	3.898	Q1

## Data Availability

The Web of Science (WoS) data can be accessed through theWoS’s official website: https://www.webofscience.com/wos/alldb/basic-search (accessed on 14 March 2022).

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
