# Peer review of "The Top 100 Most Cited Scientific Papers in the Public, Environmental & Occupational Health Category of Web of Science: A Bibliometric and Visualized Analysis"

_ijerph, 2022, doi:10.3390/ijerph19159645_

Round 1

Reviewer 1 Report

Specific comments and suggestions are included along the paper for facilitating the review process.

Author Response

Reviewer 1

First, I would like to thank the reviewers for their review work, which has undoubtedly allowed the article to be significantly improved.

Next, I proceed to specify the changes suggested by the reviewer.

  • The references have been incorporated:

  • Gusenbauer, M, Haddaway, NR. Which academic search systems are suitable for systematic reviews or meta-analyses? Evaluating retrieval qualities of Google Scholar, PubMed, and 26 other resources. Res Syn Meth. 2020; 11: 181– 217. https://doi.org/10.1002/jrsm.1378
  • Gusenbauer, M. Search where you will find most: Comparing the disciplinary coverage of 56 bibliographic databases. Scientometrics 127, 2683–2745 (2022). https://doi.org/10.1007/s11192-022-04289-7

  • Spelling errors have been corrected and words changed at the reviewer's suggestion.
  • Capital letters have been incorporated in the words suggested by the reviewer.
  • Commas have been added to numerical values
  • In line 69, the paragraph has been put aside.
  • In line 112 and 135 the numerical value has been corrected.
  • Errors in the name of the institutions in table 4 and lines 248 and 252 have been corrected.
  • All the suggestions of the bibliographical references have been modified and corrected.

Reviewer 2 Report

The first sentence tries to define 'Bibliometrics' as a 'mathematical-statistical tool'; well, this is not so; it includes mathematical and statistical tools (nota a tool), but it is not a part of Mathematics or Statistics, although it uses quantitative techniques.

Lines: 130-2: There are 632304 items, and 100 selected; thus the number of excluded were 632204 (not 532204)

Line 139: there is a reference to 'quality' of the data based on the number of publication in one time interval. What has this to do with quality?

Line 144 (and some later on): 'rs' should be defined; Spearman correlation?

Lines 144-5: p = 0.229, nor p < 0.229.

Lines 147, 156, 167 (table):  Use capital-lower case letters in the names of the journals, as they are published.

Line 176: '100 best articles' surely means '100 most cited articles'

Line 281: The claim about an upward trend is contradictory to what can be seen in figure 3; the trend could be present during the period 1989-2010, but then it dissapears in the last decade. Time series not necessarily show deterministic trends; in fact most of its components are stochastic.

Author Response

First, I would like to thank the reviewers for their review work, which has undoubtedly allowed the article to be significantly improved.

Next, I proceed to specify the changes suggested by the reviewer.

  • The first sentence tries to define 'Bibliometrics' as a 'mathematical-statistical tool'; well, this is not so; it includes mathematical and statistical tools (nota a tool), but it is not a part of Mathematics or Statistics, although it uses quantitative techniques.
    • The sentence has been redrafted based on the reviewer's input.
  • Lines: 130-2: There are 632304 items, and 100 selected; thus the number of excluded were 632204 (not 532204)
    • The mistake has been fixed.
  • Line 139: there is a reference to 'quality' of the data based on the number of publication in one time interval. What has this to do with quality?
    • The sentence has been redrafted based on the reviewer's input.
  • Line 144 (and some later on): 'rs' should be defined; Spearman correlation?
    • The mistake has been modified.
  • Lines 144-5: p = 0.229, nor p < 0.229.
    • The mistake has been modified.
  • Lines 147, 156, 167 (table):  Use capital-lower case letters in the names of the journals, as they are published.
    • bugs have been changed.
  • Line 176: '100 best articles' surely means '100 most cited articles'
    • The sentence has been redrafted based on the reviewer's input.
  • Line 281: The claim about an upward trend is contradictory to what can be seen in figure 3; the trend could be present during the period 1989-2010, but then it dissapears in the last decade. Time series not necessarily show deterministic trends; in fact most of its components are stochastic.
    • The sentence has been redrafted based on the reviewer's input